Complex hydrothermal vent microbial mat communities used to assess primer selection for targeted amplicon surveys from Kama‘ehuakanaloa Seamount

Smith Lindsey 1
http://orcid.org/0000-0001-8747-5903 Fullerton Heather 2
Moyer Craig L. 1 Craig.Moyer@wwu.edu
1 Department of Biology, Western Washington University , Bellingham, WA , United States
2 Department of Biology, College of Charleston , Charleston, SC , United States
Girguis Peter
Electronic publication date: 2024 Sep 16
Publication date: 2024
Volume: 12
Electronic Location ID: e18099
Received 2024 Apr 12; Accepted 2024 Aug 26
Copyright: © 2024 Smith et al.
Copyright year: 2024
Copyright holder: Smith et al.
License: This is an open access article distributed under the terms of the Creative Commons Attribution License, which permits unrestricted use, distribution, reproduction and adaptation in any medium and for any purpose provided that it is properly attributed. For attribution, the original author(s), title, publication source (PeerJ) and either DOI or URL of the article must be cited.
License URL: https://creativecommons.org/licenses/by/4.0/

Keywords: Hydrothermal vents, Microbial mats, Amplicon sequencing, Microbial ecology, Zetaproteobacteria, Kama‘ehuakanaloa Seamount

Funding: Fouts Foundation for the Enhancement of Student Research Experiences National Science Foundation OCE 1155756 This work was supported by the Fouts Foundation for the Enhancement of Student Research Experiences and by the National Science Foundation, award OCE 1155756 to Craig Moyer. There was no additional external funding received for this study. The funders had no role in study design, data collection and analysis, decision to publish, or preparation of the manuscript.

==============================
The microbiota of hydrothermal vents has been widely implicated in the dynamics of oceanic biogeochemical cycling. Lithotrophic organisms utilize reduced chemicals in the vent effluent for energy, which fuels carbon fixation, and their metabolic byproducts can then support higher trophic levels and high-biomass ecosystems. However, despite the important role these microorganisms play in our oceans, they are difficult to study. Most are resistant to culturing in a lab setting, so culture-independent methods are necessary to examine community composition. Targeted amplicon surveying has become the standard practice for assessing the structure and diversity of hydrothermal vent microbial communities. Here, the performance of primer pairs targeting the V3V4 and V4V5 variable regions of the SSU rRNA gene was assessed for use on environmental samples from microbial mats surrounding Kama‘ehuakanaloa Seamount, an iron-dominated hydrothermal vent system. Using the amplicon sequence variant (ASV) approach to taxonomic identification, the structure and diversity of microbial communities were elucidated, and both primer pairs generated robust data and comparable alpha diversity profiles. However, several distinct differences in community composition were identified between primer sets, including differential relative abundances of both bacterial and archaeal phyla. The primer choice was determined to be a significant driver of variation among the taxonomic profiles generated. Based on the higher quality of the raw sequences generated and on the breadth of abundant taxa found using the V4V5 primer set, it is determined as the most efficacious primer pair for whole-community surveys of microbial mats at Kama‘ehuakanaloa Seamount.

Introduction

Targeted amplicon surveys of the small subunit ribosomal RNA gene (SSU rRNA gene) are a fast and cost-effective culture-independent approach for microbial ecology studies. Since the SSU rRNA gene contains both variable and conserved regions it allows for robust taxonomic identification (McNichol et al., 2021). Additionally, this approach provides relative abundance data that distinguishes both rare and abundant taxa, which can be used to build a picture of community structure and diversity (Pascoal, Costa & Magalhães, 2021). The design and selection of optimal SSU PCR primers has been an ongoing concern and challenge because the choice of PCR primers may result in over- or under-representation of some groups of taxa in the final dataset, which may lead to inaccurate interpretations or conclusions (Abellan-Schneyder et al., 2021; Bahram et al., 2019; Baker, Smith & Cowan, 2003; Hamady & Knight, 2009; Walters et al., 2015; Wang & Qian, 2009).

Two of the most commonly used primer sets target either the V3V4 or the V4V5 region of the SSU rRNA gene. A suite of primers was evaluated in silico for amplification efficiency and 341F/785R, targeting the V3V4 region, was recommended as the most efficacious pair despite a noted tendency for overrepresentation of Alphaproteobacteria and underrepresentation of both Bacteroidetes and the important SAR11 clade of pelagic bacteria, e.g., order Pelagibacterales (Klindworth et al., 2013). In 2016, the same strong bias against Pelagibacterales was noted when using the primers adopted by a large-scale sequencing project, the Earth Microbiome Project (EMP), as well as the underrepresentation of two archaeal phyla (Parada, Needham & Fuhrman, 2016). The EMP primers (515F/806R) only amplify the V4 region of the SSU gene, as they were designed at a time when Illumina amplicon length was constrained to 2 × 75–100 bp (Caporaso et al., 2011). To increase amplification efficiency in both Bacteria and Archaea, a different previously validated reverse primer was selected, and a new degeneracy was added to the forward primer (Parada, Needham & Fuhrman, 2016; Quince et al., 2011). The new primers, referred to as 515F-Y/926R, amplify the V4V5 region of the SSU rRNA gene. Testing against marine samples, the V4V5 primers strongly amplified Pelagibacterales and recovered multiple additional taxa not covered by the V4 primers. The V4V5 primers were tested against mock communities and produced community profiles that much more closely matched the expected composition than those generated by the V4 primers in both even and staggered trials (Parada, Needham & Fuhrman, 2016).

When evaluating primer performance, it is important to compare to environmental samples rather than simply relying on mock community data (Abellan-Schneyder et al., 2021; Wear et al., 2018). This was demonstrated during the development of the V4V5 primers: the differential abundance of Pelagibacterales detected by V4V5 was less than twofold that detected by the V4 primers in mock analysis, while in the environmental samples it was between 4- and 10-fold higher (Parada, Needham & Fuhrman, 2016). Established microbial mat communities can be highly complex and that complexity cannot feasibly be replicated with synthetic community constructions (Vander Roost, Thorseth & Dahle, 2017). A comparison of primer performance among several mock communities, saliva samples, and soil samples indicated that the degree of difference in community profiles generated by different primers may scale with increasing community complexity (Soriano-Lerma et al., 2020). For this reason, the assessment of primers for use on highly complex communities, like those found in the microbial mats at Kama‘ehuakanaloa Seamount, should be conducted directly on environmental samples.

Hydrothermal vent systems represent a gradient at the interface between the oceanic crust and the deep-sea water column, presenting an ideal habitat for lithotrophic microorganisms driven by an abundance of dissolved reduced compounds. The chemistry of the venting fluid has been shown to impact the community composition of the surrounding microbial mats, with broad variation between communities at sulfur-dominated systems and those at iron-dominated systems (Fullerton et al., 2024). Ferrous iron is rapidly oxidized abiotically, drawing it out of solution and dramatically reducing its bioavailability making it a limiting micronutrient for marine phototrophic primary production (Boyd et al., 2007). However, analysis of plumes from high-temperature focused venting shows that more iron than expected stays dissolved in the water column, and can be transported in the plume for great distances into the upper ocean (Neuholz et al., 2020; Resing et al., 2015). Microbes at the vent orifices may play a direct role in iron transport and iron speciation (Wang et al., 2021a). Recent studies support these hypotheses in the dynamics of iron transport at a diffuse vent system as well, and when comparing iron speciation from different types of vents, diffuse vent flow hosts the highest proportion of labile iron complexed with organic material (Lough et al., 2019; Wang et al., 2021b). Therefore, diffuse vent systems that host microbial mats and release fluid high in dissolved iron, like those observed at Kama‘ehuakanaloa Seamount, are of particular importance (Wheat et al., 2000; Glazer & Rouxel, 2009). The vast majority of the organisms living in the microbial mats surrounding the vents are not yet culturable, so culture-independent methods are necessary to study these communities and better elucidate their impact on global biogeochemical cycles (Emerson & Moyer, 2010; Duchinski et al., 2019).

Researchers commonly choose either the V3V4 or V4V5 region to amplify in hydrothermal vent-associated microbial mat diversity surveys (Astorch-Cardona et al., 2023; Duchinski et al., 2019; Hager et al., 2017; Ramírez et al., 2021; Scott, Glazer & Emerson, 2017; Speth et al., 2022; Stromecki et al., 2022). In this study, primer pairs for the V3V4 region (Klindworth et al., 2013) and the V4V5 region (Parada, Needham & Fuhrman, 2016) were used to amplify the SSU rRNA gene from microbial mats around the iron-dominated diffuse hydrothermal vents at Kama‘ehuakanaloa Seamount to determine if significant differences in community composition or structure would result. A deep sequencing effort of these complex communities was undertaken for both primer sets. Several distinct differences in composition between datasets were revealed in both the bacterial and archaeal domains. A comparison of overall taxonomic profiles and relative abundance data reveals that both primer sets generate robust data, but that primer choice is significant in differences among the datasets associated with iron-dominated microbial mat amplicon sequencing at Kama‘ehuakanaloa Seamount. Portions of this text were previously published as part of a thesis (Smith, 2023).

Materials and Methods

Sample collection

Samples were collected from Kama‘ehuakanaloa Seamount (formerly known as Lō‘ihi Seamount), an active undersea volcano at the leading edge of the Hawaiian hot spot (Clague et al., 2019). Four locations around the seamount were selected for sampling: Pohaku, at the southernmost edge of the caldera; Lohiau, on a small outcrop on the northern lip of the caldera; Hiolo North, within the eastern side of the caldera; and Hiolo South, just south of Hiolo North (Fig. S1). Metadata for each pair suction samples collected at each of the four locations is shown in Table 1. All venting locations emitted fluid that ranged from 20–48 °C and all were surrounded by abundant microbial mat (Scott, Glazer & Emerson, 2017; Fullerton et al., 2017).

Table 1 Information on sample collection location and sample name used in analysis.

Collection ID	Sample name	Primer	Location	Latitude	Longitude	Depth (m)	
J2-674-Green	Bact674Green	V3V4	Pohaku	18.9013	−155.2582	1,179	
Uni674Green	V4V5	
J2-674-Blue	Bact674Blue	V3V4	Pohaku	18.9015	−155.2582	1,182	
Uni674Blue	V4V5	
J2-677-Black	Bact677Black	V3V4	Hiolo South	18.9055	−155.2570	1,270	
Uni677Black	V4V5	
J2-675-Black	Bact675Black	V3V4	Hiolo South	18.9056	−155.2570	1,272	
Uni675Black	V4V5	
J2-677-Green	Bact677Green	V3V4	Hiolo North	18.9064	−155.2570	1,300	
Uni677Green	V4V5	
J2-677-Blue	Bact677Blue	V3V4	Hiolo North	18.9065	−155.2569	1,300	
Uni677Blue	V4V5	
J2-676-Black	Bact676Black	V3V4	Lohiau	18.9089	−155.2577	1,175	
Uni676Black	V4V5	
J2-676-Green	Bact676Green	V3V4	Lohiau	18.9089	−155.2577	1,175	
Uni676Green	V4V5	

Microbial mats were collected in March 2013 on the R/V Thomas G. Thompson cruise TN293 with ROV Jason II. Eight bulk mat samples were collected with an impeller-driven suction device into chambers using a double layer 202 µm Nytex catchment barrier. Samples were homogenized and then aseptically transferred to sterile 50 ml centrifuge tubes inside a cold room, preserved in RNAlater (ThermoFisher Scientific, Waltham, MA, USA), and stored at −80 °C until DNA extractions were performed in the lab.

DNA extraction, amplification, and sequencing

Genomic DNA was extracted from each sample in triplicate using the FastDNA SPIN Kit for Soil (MP Biomedicals, Santa Ana, CA, USA) after fluid removal, as previously described (Hager et al., 2017; Jesser et al., 2015). Extracted DNA was quantified using a Qubit 2.0 fluorometer (Thermo Fisher Scientific, Waltham, MA, USA).

Each sample collection was PCR amplified in quintuplicate using each primer set, for sixteen pooled amplicon sets across the eight sample collections. The V3V4 region of the SSU rRNA gene was amplified using 341F (5′-CCTACGGGNGGCWGCAG-3′) and 785R (5′-GGACTACHVGGGTATCTAATCC-3′), and the V4V5 region was amplified using 515F-Y (5′-GTGYCAGCMGCCGCGGTAA-3′) and 926R (5′-CCGYCAATTYMTTTRAGTTT-3′), then amplicons were cleaned and indexed according to Illumina MiSeq best-practices (Illumina, 2013). Amplicon libraries were quantified with a Qubit 2.0 fluorometer, and sequencing was performed with v3 chemistry on an Illumina MiSeq for 2 × 300 bp reads according to standard protocol (Illumina, 2013).

Sequence processing

Raw FASTQ files were first primer-trimmed with Cutadapt (Martin, 2011) using the linked-adapter protocol with the complete primer sequences; untrimmed sequences were discarded. Trimmed FASTQ files were then imported to R and each primer dataset was run independently using the DADA2 analysis pipeline, following established best practices as previously described (Callahan et al., 2016). Reverse primers were truncated at the position where the average quality score dropped below 20. The core inference algorithm was applied using pseudopooling. Taxonomy was assigned to the genus level using the DECIPHER R package with the IDTAXA command (Murali, Bhargava & Wright, 2018; Quast et al., 2013) aligning to the SILVA reference database v138 (Yilmaz et al., 2014). Finally, ASVs assigned to the class Zetaproteobacteria were analyzed by ZetaHunter (https://github.com/mooreryan/ZetaHunter) for classification to previously defined Zetaproteobacterial OTUs (e.g., zOTUs) (McAllister, Moore & Chan, 2018).

ASVs from each primer pair that were unidentified at the domain level were removed from the analysis. To combine the two datasets, ASVs identified to at least the domain level were then iteratively clustered down to the genus level such that ASVs binned inside a single genus were condensed into a single entry in the taxonomic table, ASVs that were unclassified at the genus level were clustered by family, and so on. Abundances of each ASV in each cluster were also condensed per sample. This compressed taxonomy is subsequently referred to as the “condensed taxa” and allows for direct comparisons between primer sets.

Diversity and statistical analysis

Rarefactions were plotted for the non-condensed ASV data per primer set using tidyverse and R version 4.1.3 (Wickham et al., 2019; R Core Team, 2022). Non-condensed ASV data were also used to generate bar plots and relative abundance pie charts for domain-, phylum-, class- and genus-level taxonomic rankings in each primer set. Alpha diversity metrics (Chao1 Richness, Shannon Diversity, Simpson Evenness) were calculated for non-condensed ASV data in each sample using the microbiome package. Prior to analysis, data in each metric were checked for normal distribution using the Shapiro-Wilke test. Differences between primer sets were assessed for statistical significance in each metric for the combined sample data: Analysis of Variance was used for Shannon Diversity and Chao1 Richness and Kruskal-Wallis was used for Simpson Evenness.

The condensed taxa sample data matrices were read into phyloseq (McMurdie & Holmes, 2013) and then transformed using variance stabilizing transformation (VST) with package DESeq2 (Love, Huber & Anders, 2014). The Euclidean distance matrix was calculated for the transformed data. Normalized data were PCoA plotted with ggplot2 (Wickham, 2016). The vegan package was used to analyze the distance matrix; first, beta dispersion by primer choice was validated as non-significant using betadisper2, and then PERMANOVA via adonis2 was performed using primer choice as the variable (Oksanen et al., 2022). PERMANOVA to assess location as a driver of variation between primer sets could not be conducted because the beta dispersion homogeneity assumption was not met for that variable.

To further examine statistically significant differences in community composition between primer sets, indicator species analysis was performed on the condensed taxa matrix. Using the multipatt command from the indicspecies package, the point biserial correlation coefficient was calculated for each taxon at minimum of 95% confidence with 104 permutations (Cáceres & Legendre, 2009).

A repository of all scripts used in data processing for this project can be accessed at https://github.com/benedil/Iron-Mat-Primer-Compare/tree/main.

Results

Sequence processing

The chosen sequencing approach resulted in sixteen community profiles representing eight collections each, divided by the two primer pairs. Amplicon processing through the DADA2 pipeline generated a total of 7,506 ASVs in 1,952,073 reads from the eight communities amplified by the V3V4 primers. This output represents 36.92% of the total input reads, since 54.61% to 75.11% of reads were removed from each single community during the DADA2 pipeline (i.e., filtering, denoising, merging, and chimera removal; Table S1). After removing ASVs that were unclassified at the domain level, the V3V4 dataset comprised 4,140 ASVs in 1,558,201 reads, meaning that 20.18% of filtered sequences could not be identified to at least the domain level. DADA2 processing of the eight communities amplified by the V4V5 primers generated a total of 5,130 ASVs in 2,950,958 reads. This translates to 47.83% of the total input reads, with 48.43% to 57.27% of reads removed from a single community during data processing with the DADA2 pipeline (Table S2). After removing ASVs that were unclassified at the domain level, the V4V5 dataset comprised 3,288 ASVs in 2,227,954 reads, meaning that 24.50% of filtered reads could be not identified to at least the domain level.

Community profiling

For the V3V4 primer set, rarefaction curves had a mean depth of 244,009 reads per community with a minimum of 130,311 and a maximum of 323,532. With the V4V5 primer set, rarefaction curves had a mean depth of 335,119 reads per community with a minimum of 316,860 and a maximum of 365,477. Plateau was reached in all communities from both primer sets when using rarefaction (Fig. S2).

After calculating alpha diversity metrics, no significant difference between primer sets in Chao1 Richness was found using Kruskal-Wallis and no significant difference in Shannon Diversity or Simpson Evenness was found using Analysis of Variance (Fig. 1). PERMANOVA analysis supported the hypothesis that the primer set was a significant driver of the differences observed among the taxonomic profiles (R2 = 0.25, P = 0.001). PCoA of the normalized data explained 51.6% of the total variance in both principal coordinates and showed distinct clusters associated with each primer set at the 95% confidence interval (Fig. 2).

Figure 1 Alpha diversity per primer set.

(A) Chao1 Richness, (B) Shannon Diversity, and (C) Simpson Evenness estimates for each microbial mat community. Different primer sets are represented by different colors. Significance was assessed using analysis of variance for normally distributed data (Shannon and Chao1) or Kruskal-Wallis for non-normally distributed data (Simpson).

Figure 2 Principal coordinate analysis of microbial mat communities by primer set, with ellipses indicative of a 95% confidence interval.

Domain Archaea

The V3V4 primer set generated 62 Archaeal ASVs spread over six phyla plus another group that was unclassified except at the domain level. Twenty-six ASVs were identified as Crenarchaeota, 13 as Thermoplasmatota, eight each as Hydrothermarchaeota and Halobacterota, three as Nanoarchaeota, three were unclassified at the phylum level, and a single ASV identified as Euryarchaeota. Phylum Asgardarchaeota was unrepresented in the V3V4 dataset (Fig. 3A). By relative abundance in domain Archaea, Thermoplasmatota was the most abundant phylum with 41.79% of reads, followed by Crenarchaeota with 22.81% and Hydrothermarchaeota with 20.49% (Fig. 3B).

Figure 3 Diversity in domain Archaea captured by each primer set.

(A) As described via ASVs per phylum and as described via phylum relative abundance for (B) primer set V3V4 and (C) primer set V4V5. In (B) and (C) legend, †denotes taxa present in V4V5 but absent in V3V4, ‡denotes taxa present in V3V4 but absent in V4V5, °denotes taxa that differ in relative abundance by at least one order of magnitude between primer sets. NB: in (B), the non-visible 0.06% slice represents the phylum Euryarchaeota.

The V4V5 primer set generated approximately two-thirds again as many Archaeal ASVs as the V3V4 primer set, with 103 ASVs spread over six phyla plus another group that was unclassified except at the domain level. Crenarchaeota was the dominant phylum in terms of both the number of representative ASVs, with 53, and in relative abundance within the domain Archaea (63.42% of archaeal reads; Figs. 3A and 3C). Phylum Thermoplasmatota was represented by 19 ASVs, 11 were classified as Nanoarchaeota (the next greatest by relative abundance, with 10.26% of reads), seven as Hydrothermarchaeota, four as Halobacterota, and three were unclassified at the phylum level. All but Crenarchaeota and Nanoarchaeota composed less than 10.00% of total archaeal reads (Fig. 3C). In contrast to the V3V4 samples, phylum Asgardarchaeota was represented by seven ASVs and phylum Euryarchaeota was unrepresented (Fig. 3A).

Of the eight identified divisions in the domain Archaea (inclusive of unclassified phyla), six differed by at least one order of magnitude in relative abundance between primer sets. Halobacterota and Crenarchaeota abundances were within the same order of magnitude between primers, although V4V5 had more than three times as many reads identified as Crenarchaeota than did V3V4. Both primers detected one phylum unique to their respective dataset. Phylum Euryarchaeota, unique to V3V4, was represented by a single ASV that accounted for 0.06% relative abundance. Meanwhile, phylum Asgardarchaeota, unique to V4V5, was represented by seven ASVs that accounted for 0.95% relative abundance (Figs. 3A–3C). Only Halobacterota abundances were similar between primer sets, despite differences in representative ASVs (Figs. 3A–3C).

Domain Bacteria

There were 4,078 ASVs recovered from the V3V4 primer set that classified to the domain Bacteria, belonging to 49 phyla (Fig. 4A). NB: Phylum Pseudomonadota was referred to as phylum Proteobacteria at the time of this analysis, so all instances of the term Proteobacteria in this work should be considered as synonymous with Pseudomonadota (Oren & Garrity, 2021). The phyla with the greatest number of ASVs were Proteobacteria with 1,322, followed by Planctomycetota with 498 ASVs, Bacteroidota with 262 ASVs, Patescibacteria (sometimes called Candidate Phyla Radiation, or CPR) with 206 ASVs, Verrucomicrobiota with 146 ASVs, and Chloroflexi with 135 ASVs. There were 709 ASVs unclassified below the domain level. By relative abundance, 12 phyla made up at least 1% of the reads, respectively (Fig. 4B). Proteobacteria made up fully half of the bacterial reads (50.73%), with the next most abundant phyla being Planctomycetota (8.11%), Bacteroidota (7.17%), Patescibacteria (6.32%), and the uncharacterized DTB120 (4.83%). Phyla with relative abundance between 0.99% and 0.10%, were led by Gemmatimonadota, Acidobacteriota, Zixibacteria, Verrucomicrobiota, and Bdellovibrionota. The lowest relative abundance phyla (those under 0.10% abundance) were the most numerous, with 22 different representatives, but their 77 ASVs altogether accounted for just 0.28% of total bacterial reads.

Figure 4 Diversity in domain Bacteria captured by each primer set.

(A) As described via ASVs per phylum for phyla representing greater than 1% relative abundance in either primer set and via phylum relative abundance for (B) primer set V3V4 and (C) primer set V4V5. In (B) and (C) legend, °denotes taxa that differ in relative abundance by at least one order of magnitude between primers.

The V4V5 primer set generated 3,185 ASVs classified to domain Bacteria, representing 55 phyla (Fig. 4A). The phyla with the most representative ASVs were Proteobacteria with 979 ASVs, followed by Planctomycetota with 447 ASVs, Bacteroidota with 289 ASVs, Chloroflexi with 128 ASVs, and Verrucomicrobiota with 121 ASVs. There were 393 ASVs unclassified below the domain level. By relative abundance, eight phyla contributed greater than 1.00%, respectively (Fig. 4C). Proteobacteria made up more than half of all reads (58.44%), followed by Bacteroidota (9.80%), Planctomycetota (5.61%), Nitrospirota (5.42%), and DTB120 (4.77%). The top five most abundant phyla between 0.10% and 1.00% relative abundance were MBNT15, Gemmatimonadota, Zixibacteria, Patescibacteria, and Bdellovibrionota. There were 28 phyla with relative abundances below 0.10%, whose 103 ASVs comprised 0.37% of total bacterial reads.

There were several dissimilarities between the two bacterial taxonomic profiles. Two phyla were unique to the V3V4 dataset (GAL15 and Poribacteria) and eight phyla were unique to the V4V5 dataset (Caldatribacteriota, CK-2C2-2, Deferrisomatota, Edwardsbacteria, Elusimicrobiota, FCPU426, Latescibacterota, and TA06), although all unique phyla comprised less than 1.00% relative abundance in their respective datasets. Four phyla with greater than 0.10% relative abundance in at least one primer differed in that abundance by at least one order of magnitude: Patescibacteria (V3V4: 6.32%, V4V5: 0.50%), Campylobacterota (V3V4: 2.19%, V4V5: 0.33%), Acetothermia (V3V4: 0.34%, V4V5: 0.03%), and Deinococcota (V3V4: 0.002%, V4V5: 0.16%). Patescibacteria and Campylobacterota also differed more than any other taxa in numbers of representative ASVs between primer sets, with V3V4 having 3.5-fold more Patescibacteria ASVs and 3-fold more Campylobacterota ASVs. There were two additional taxonomic groupings with notable variation in ASVs between primers: phylum Proteobacteria and taxa unclassified below domain ranking (Fig. 4A). While those two groupings contained 659 ASVs unique to the V3V4 dataset (approximately 75% of the total number of ASVs unique to V3V4), all the unique ASVs were present in very low abundance (Figs. 4B and 4C). However, four out of the top five most abundant phyla in each primer set were the same and their numbers of representative ASVs were within the same order of magnitude. The one exception to this trend was in phylum Patescibacteria.

Phyla Proteobacteria (i.e., Pseudomonadota) and Campylobacterota

The V3V4 primer set generated 1,322 ASVs identified to phylum Proteobacteria and 83 phylum Campylobacterota ASVs. Within the two phyla, class Zetaproteobacteria was represented by 266 ASVs and made up 61.18% of total reads (Figs. 5A and 5B). In contrast, class Gammaproteobacteria was the most diverse with 623 ASVs, but accounted for just 21.39% of total reads. Next greatest by abundance was class Alphaproteobacteria with 350 ASVs constituting 13.14% of reads, followed by a mere 0.16% of reads consisting of 83 ASVs unclassified below phylum level in Proteobacteria (Fig. 5A). All phylum Campylobacterota ASVs were also classified as class Campylobacteria and composed 4.13% of reads.

Figure 5 Diversity in phylum Proteobacteria captured by each primer set.

(A) As described via ASVs per class and via class relative abundance for (B) primer set V3V4 and (C) primer set V4V5. In (B) and (C) legend, °denotes taxa that differ in relative abundance by at least one order of magnitude between primers. NB: Class Campylobacteria in phylum Campylobacterota is included in the analysis of phylum Proteobacteria here since it was formerly class Epsilonproteobacteria.

Phylum Proteobacteria had 979 ASVs in the V4V5 dataset and phylum Campylobacterota had just 27 ASVs. Class Zetaproteobacteria repeated the trend observed in the V3V4 dataset but to an even greater extent: by relative abundance, Zetaproteobacteria were in the definitive majority with 74.98% of total reads but were represented by only 89 ASVs (Figs. 5A and 5C). Gammaproteobacteria profiles were also similar to V3V4, with an extensive collection of 531 ASVs comprising only 15.54% of total reads within the two phyla. Class Alphaproteobacteria had nearly the same number of ASVs as were present in the V3V4 dataset, but they made up only about two-thirds as much of the V3V4 relative abundance. Only 26 ASVs remained unclassified below the phylum level in Proteobacteria, and they accounted for 0.35% of total reads. Campylobacterota made up only 0.56% of total reads.

Two interesting differences among the taxonomic profiles within these data are the relative abundances of both Campylobacterota and Zetaproteobacteria. The relative abundances of Campylobacterota differed by an order of magnitude, with V3V4 having the greater representation. Zetaproteobacteria abundances differed to a lesser degree, with V4V5 containing approximately 15% relatively more Zetaproteobacteria reads. The relative discrepancy of Zetaproteobacteria reads at the class level translated to a difference of more than 10% in the relative abundance of Zetaproteobacteria between the two datasets as a whole; Zetaproteobacteria accounted for 32.28% of total identified reads in V3V4 and 45.54% of total identified reads in V4V5.

Indicator species analysis

Indicator species analysis revealed that of 916 representatives in the condensed taxa, 123 were significantly correlated with one primer pair or the other: 50 taxa correlated with the V3V4 primer set (Table S3), while 73 taxa correlated with the V4V5 primer set (Table S4). The V3V4-correlated taxa were distributed within twelve bacterial phyla; phylum Patescibacteria encompassed 18 taxa, the greatest number of indicator species represented by a single phylum in the V3V4 dataset. The most abundant classes were largely within phylum Patescibacteria, but classes Gammaproteobacteria and Planctomycetes were also well represented (Fig. 6A). By total relative abundance in the V3V4 dataset, the 50 correlated taxa accounted for 8.30% of the reads, with only 23 taxa in ten phyla each making up greater than 0.10% relative abundance. In contrast, the 73 V4V5-correlated taxa accounted for 49.30% of total V4V5 reads. The majority of that relative abundance was explained by the presence of a Zetaproteobacteria, which was the single most highly abundant taxon in the condensed taxa matrix. Thirteen phyla and one group unclassified at the phylum level held 26 taxa that each made up at least 0.10% relative abundance (Fig. 6B). The V4V5 primers correlated significantly with 25 phyla in total: six archaeal and nineteen bacterial. Phylum Proteobacteria had the greatest number of representatives in V4V5, with 21, although only five were in higher abundance. Bacterial classes Bacteroidia, Ignavibacteria, and the uncharacterized Nitrospirota class BMS9AB35 were well represented in the analysis, as well as the archaeal class Nitrososphaeria. While some taxa that correlated with one primer pair were completely absent in the other pair, others, like the Zetaproteobacteria, were found in the datasets of both primer pairs; the two sets of correlated taxa were not mutually exclusive between primers.

Figure 6 Point biserial correlation analysis of condensed taxa.

(A) With V3V4 primer set and (B) V4V5 primer set. All taxa are present in at least 0.1% relative abundance in their respective datasets. Class bubbles are colored by phylum and sized by percent relative abundance within each dataset.

Class Zetaproteobacteria

There were 26 previously established Zetaproteobacteria OTUs (zOTUs) found with the V3V4 primer set (Fig. 7A). By far the most abundant was zOTU2, with 56.46% of the total zOTU reads (Fig. 7B). zOTU1 followed with 15.76%, then zOTU4 (8.20%), zOTU10 (7.11%), and zOTU14 (2.09%). The majority of the remaining zOTUs were represented by less than 1.00% of the zOTU reads (Fig. 7C). Additionally, 75 sequences were classified as “NewZetaOTUs”, all of which were found in very low abundance, each with fewer than 20 total reads.

Figure 7 Diversity in class Zetaproteobacteria captured by each primer set.

(A) As described via reads per zOTU designation. NB: the break in the y-axis represents a change in scale with respect to reads per zOTU. Zetaproteobacteria diversity is also described per primer set for (B) primer set V3V4 and (C) primer set V4V5 via high-abundance taxa, (i.e., greater than 104 reads per taxon), and for (D) primer set V3V4 and (E) primer set V4V5 via low-abundance taxa, (i.e., fewer than 104 reads per taxon). In (D) and (E) legend, †denotes taxa present in V4V5 but absent in V3V4, ‡denotes taxa present in V3V4 but absent in V4V5, and ° denotes taxa that differed in relative abundance by at least one order of magnitude between primer sets.

The V4V5 primers retrieved 24 previously established zOTUs (Fig. 7A). As with the V3V4 dataset, zOTU2 was the most dominant taxon with 55.99% of the zOTU reads (Fig. 7D). zOTU1 was next in abundance with 9.45%, then zOTU14 (9.56%), zOTU10 (6.41%), and zOTU4 (4.63%). Similar to the V3V4 dataset, the majority of identified zOTUs were present in less than 1% relative abundance (Fig. 7E). Only 16 “NewZetaOTUs” were found by the V4V5 primers, all in very low abundance.

The V4V5 primers retrieved nearly twice as many total Zetaproteobacteria reads as did V3V4 (990,865 in V4V5 compared to 502,938 in V3V4). Both V3V4 and V4V5 identified the same eight zOTUs as most abundant and although the absolute number of reads in each zOTU, respectively, differed by up to nine-fold (zOTU14; Fig. 7A), when comparing the data by relative abundance the profiles were quite similar. This trend in absolute versus relative abundance is also observed in the lower abundance zOTUs (Figs. 7A, 7C and 7E). However, there are several exceptions to the similarities in the lower abundance zOTUs. Two zOTUs differed by at least one order of magnitude in relative abundance: zOTU18 (V3V4: 0.074%, V4V5: 1.047%) and zOTU26 (V3V4: 0.029%, V4V5: 0.001%). Additionally, there were zOTUs present in one dataset that were completely absent in the other: zOTU54 and zOTU12 were not found using the V3V4 primers, while zOTUs 52, 31, 23, and 40 were missing from the V4V5 data. All unique zOTUs, regardless of the dataset, were represented by less than 1% relative abundance, and all but two (zOTU52 in V3V4 and zOTU54 in V4V5) were represented by less than 0.1% relative abundance (Figs. 7D and 7E).

Discussion

Primer selection has previously been evaluated as a source of potential bias in microbial ecology studies of various complex communities and primer selection has been determined to be significant as a driver of variation in habitats such as the human microbiome, soils, terrestrial mineral deposits, the pelagic water column, marine subsurface sediment, and sea ice (Bahram et al., 2019; Fadeev et al., 2021; Hathaway et al., 2021; McNichol et al., 2021; Nearing, Comeau & Langille, 2021; Pollock et al., 2018; Rajeev et al., 2020; Walters et al., 2015). Targeted amplicon surveys of the iron-dominated microbial mats at Kama‘ehuakanaloa Seamount have been an important analytical tool for assessing diversity and community structure, but there is no consensus on the optimal primer pair to use for such surveys. The V4V5 primers used here show greater promise than the V3V4 primers for analysis of hydrothermal vent-associated iron-dominated microbial mats at Kama‘ehuakanaloa Seamount, and these results may have implications for the study of other complex microbial communities.

The protocols used in this study were chosen after a careful review of the current methodological literature. Sequencing depth has been shown to influence measures of alpha diversity in microbial ecology studies, both in terms of richness and evenness (Bardenhorst et al., 2022; Ramakodi, 2021; Reese & Dunn, 2018). In undertaking a deep sequencing effort, biases in alpha diversity measures have been minimized, therefore assuring the robustness of the generated datasets. The lack of statistically significant differences in alpha diversity profiles between primer sets confirms the suitability of downstream comparisons of taxonomic profiles.

Differences in taxonomic profiles were apparent at each ranking examined, which is consistent with the findings of other studies that compared these two primer pairs using environmental samples. Primer choice exhibited a detectible effect in a study of terrestrial soil microbial communities at both the phylum and genus levels, with the difference between V3V4 and V4V5 more pronounced than when comparing V4V5 to other regions (Soriano-Lerma et al., 2020). In that same study, V4V5 was also able to detect variation in composition across the broadest range of community complexity as measured by soil development, compared to other hypervariable regions. However, when they compared the less complex microbial community of human saliva, the effect of primer choice was weaker. In an examination of various arctic microbial communities, taxa at both the class and family level were differentially abundant based on V3V4 or V4V5 primer choice (Fadeev et al., 2021). The differences among community profiles increased in amplitude with increasing community complexity, with the greatest differential present in the marine sediment community. Interestingly, even though 14% of the total lineages found with V3V4 in that study were absent from the V4V5 profile, those lineages accounted for less than 1% of the total sequences in the V3V4 dataset. This same trend was observed in the current study: while V3V4 produced a greater number of taxa, those taxa that were unique to V3V4 were almost exclusively of very low abundance. The trend is continued when the synthesis of ASVs per taxon and the relative abundance of those taxa are compared between primer sets. Despite the relative abundance data looking similar for each primer set, that abundance is divided into more ASVs in the V3V4 dataset; there are more ASVs sharing each piece of the pie, indicating that there are fewer representatives per taxon than in the V4V5 dataset. Moreover, the relative abundance data are validated as independent of sequencing effort by the plateaus reached in the rarefaction curves, which show that increased sequencing depth would not reveal additional diversity.

Calculation of point biserial correlation coefficients for the condensed taxa helped provide greater context for the differentially abundant taxa in each primer set. While correlational analysis is typically used in more direct ecology-focused studies, in this case, primer choice can be considered the dichotomous variable. An earlier primer comparison study also found distinct taxa associated with particular primer pairs using a correlational analysis (Fischer et al., 2016). Given the observed greater number of taxa present in the V3V4 dataset, it was surprising that the taxa correlated to V3V4 were fewer in both the number of taxa and number of recruited phyla than in the V4V5-correlated taxa. While the taxa themselves were largely unsurprising, it was gratifying to find statistical rigor hidden in observed trends. The inclusion of the Zetaproteobacteria taxon in the V4V5-correlated taxa was particularly interesting, given its dominance in both datasets. However, there were also some unexpected outcomes. Despite the noted tendency of V3V4 to capture more Campylobacterota ASVs in greater abundance than V4V5, Campylobacterota were poorly represented in the V3V4-correlated taxa.

The high-resolution taxonomic analysis of class Zetaproteobacteria, a clade of iron-oxidizing bacteria, is of particular importance concerning the study of biogeochemistry at iron-dominated diffuse hydrothermal vents. An alignment and analysis of the sequences generated by both primer sets should be undertaken to determine the loci of variations that convey taxonomic identity and to determine if the few zOTU holes in the V4V5 dataset can be filled. As part of their iron metabolism, Zetaproteobacteria produce extracellular structures composed of Fe(III) waste products complexed with carboxyl-rich polysaccharides (Chan et al., 2011). Iron that has been complexed with organic components remains in the dissolved fraction of total iron in the water column much more efficiently than iron complexed with inorganic colloids, and spectral and micro-X-ray fluorescence analysis of diffuse plume-associated iron-bearing particles revealed them to be rich in Fe(III), carbon, and oxygen (Lough et al., 2019). This profile is consistent with that of the biomineral stalks produced by Zetaproteobacteria (Chan et al., 2011). Iron complexation with exopolysaccharides has been shown to enhance bioavailability to marine phytoplankton, which have been estimated to contribute approximately half of the planetary primary production (Field et al., 1998; Hassler et al., 2011, 2015). The ability to accurately catalog the zetaproteobacterial diversity at diffuse vents is a necessary step in the evaluation of how microbial activity in these systems may be shaping primary productivity in the surrounding oceans.

Across the breadth of taxonomic rankings, the V4V5 primer set tested here revealed the greatest range of the most abundant taxa present in the study system, particularly due to the inclusion of a much broader assortment of domain Archaea. The V3V4 primer set presented a slightly different picture, with often significantly fewer representatives of a larger overall community. However, caveats exist for both primer sets. When assessing experimental methodologies, the question at hand must always have the greatest consideration. The methodology used here gives insight to the communities at Kama‘ehuakanaloa Seamount, but it may also be used in future studies to provide additional context to primer comparisons in other environments. The underrepresentation of archaea is quite a large hindrance to the use of V3V4 in the comprehensive surveying of microbial communities. While examining the performance of primers targeting archaea specifically, the V3V4 region has been shown to have a high incidence of unspecific amplification (Fischer et al., 2016). Conversely, the underrepresentation of the Campylobacterota in the V4V5 dataset may be problematic for some research questions, particularly in ecosystems rich in sulfur (Dick, 2019). A recent study used the V3V4 primers in an examination of the potential effects of climate change on the composition and ecological function of salt marsh microbial mats; the analysis presented here may suggest a reassessment of the completeness of their evaluations (Mazière et al., 2023). The impact of environmental factors on hydrothermal microbial community development in the sulfur-rich Guaymas Basin field was recently assessed using V4V5 primers (Ramírez et al., 2021); the current study suggests that the use of those primers might limit their findings.

Choosing PCR primers for environmental surveys can be difficult. While employing multiple sequencing efforts using different primers to capture the diversity of different taxa may be possible, the cost-effectiveness of using a single primer pair to capture diversity may also be desirable. The V4V5 primers detected a greater number of archaeal taxa and a greater range of bacterial phyla in Kama‘ehuakanaloa Seamount microbial communities. Given that the V4V5 primers were able to amplify some of the underrepresented taxa rather than missing them entirely, reevaluating the existing 515F-Y and 926R primers for possible additional degeneracies seems the more plausible scenario for bridging the taxonomic gaps between the two primer sets. Particular attention should be given to determining the cause of the “blind spots” within the spectrum of zOTUs captured by V4V5. To balance the inclusion of additional degeneracies with the resulting drop in PCR efficiency, primers can now be validated to select only combinations of oligonucleotide mixtures that have been identified within the reference database or in the natural environment. Additionally, using a combination of metagenomic and amplicon approaches, several loci in the V4V5 forward primer have already been identified where added degeneracies would resolve the majority of missing Patescibacteria (McNichol et al., 2021). Work to improve the V4V5 primers evaluated here should continue along this trajectory.

Conclusions

Based on the initial higher quality of the raw sequences and the composition of the taxonomic profiles produced by this study, the V4V5 region of the SSU rRNA gene is the more efficacious area for targeted amplicon surveys of iron-dominated microbial mats at Kama‘ehuakanaloa Seamount. However, the results of this study apply particularly to this study system. The high specificity of microbiota per environment may preclude direct between-environment comparisons of primer performance. It is only through repeated per-environment examinations of primer performance that sturdy foundations for future studies can be built. The methodology tested here can be utilized in future studies at other iron-dominated venting sites, to continue the validation process. Furthermore, sequence analysis is only as good as the reference databases used, so continued expansion of high-quality curated reference sequence databases is crucial. As reference databases improve, PCR primer design will improve. As primer design improves, not only do the established amplicon protocols improve, but developing methodologies like hybrid amplicon-shotgun metagenomic surveys may gain greater rigor.

Supplemental Information

Supplemental Information 1 Read tracking through the DADA2 pipeline for the V3V4 samples.

Supplemental Information 2 Read tracking through the DADA2 pipeline for the V4V5 samples.

Supplemental Information 3 Representatives of the condensed taxa that significantly correlated with the V3V4 primer set. “rpb” refers to the point biserial correlation coefficient.

Supplemental Information 4 Representatives of the condensed taxa that significantly correlated with the V4V5 primer set. “rpb” refers to the point biserial correlation coefficient.

Supplemental Information 5 Sampling locations.

Bathymetric map (high resolution at <2 m) of sampling sites in and near Pele’s Pit caldera on the summit of Kama‘ehuakanaloa Seamount, Hawai’i. Precise marker locations include Pohaku (Marker 57), Lohiau (Marker 2), Hiolo North (Markers 36, 39 and 31), and Hiolo South (Markers 34, 38 and Ku’kulu). Courtesy of Susan Merle, NOAA EOI/OSU (Clague et al., 2019).

Supplemental Information 6 Rarefaction curves.

Rarefaction curves for each microbial mat community, colored by sample location for both primer sets. Vertical lines denote the minimum read depth. NB: The difference in the x-axis scales.

We ardently thank the captain and crew of the R/V Thomas G. Thompson and the operation team for ROV Jason II for their assistance with sample collection during oceanographic research cruise TN293.

Additional Information and Declarations

Competing Interests

Author Contributions

Data Availability

Craig Moyer is an Academic Editor for PeerJ.

Lindsey Smith conceived and designed the experiments, performed the experiments, analyzed the data, prepared figures and/or tables, authored or reviewed drafts of the article, and approved the final draft.

Heather Fullerton conceived and designed the experiments, analyzed the data, authored or reviewed drafts of the article, and approved the final draft.

Craig L. Moyer conceived and designed the experiments, analyzed the data, authored or reviewed drafts of the article, and approved the final draft.

The following information was supplied regarding data availability:

The sequence data are available through NCBI Sequence Read Archive (SRA) BioProject number PRJNA1080611.

A repository of all scripts used for this project is available at GitHub and Zenodo:

- https://github.com/benedil/Iron-Mat-Primer-Compare/tree/main

- benedil. (2024). benedil/Iron-Mat-Primer-Compare: Kama‘ehuakanaloa Iron Microbial Mat Primer Comparison Study (v1.0). Zenodo. https://doi.org/10.5281/zenodo.13382128.

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
