# Peer review of "Complex hydrothermal vent microbial mat communities used to assess primer selection for targeted amplicon surveys from Kama‘ehuakanaloa Seamount"

_PeerJ, doi:10.7717/peerj.18099_

## Round 0.1 · original submission · Major Revisions

In addition to the comments provided by the peer reviewers, I would add that the manuscript could be improved by reducing some of the broad generalizations to more focused statements/conclusions. For example, it is appropriate to describe this study as an assessment of primers for use in iron-rich systems such as the study site. You might also note that community composition varies widely across hydrothermal systems, e.g. iron-rich vs. sulfur-rich vs. serpentenizing systems. You might also consider noting that other efforts have looked at primers for sulfur-rich systems but -relatively speaking- less work has focused on iron-rich systems.

Reviewer 1 ·

Basic reporting

The result section describes in detail what is shown (or could efficiently been shown) if Figures / Tables.
This makes the results section very heavy to read. The text lacks clarity regarding what the authors want to achieve and how the data support their claims.

Experimental design

16S rRNA amplicon analyses is a well established method on molecular microbial ecology.
However, using samples from one site to investigate the general validity of primer pairs does not seem like a valid approach (see below).

Validity of the findings

The major aim of the paper seems to be to evaluate two primer sets. To do this, samples from four locations at one Seamount were analyzed by comparing results from the two primer sets.
It is well known that there are biases in 16S rRNA amplicon sequencing and the different primers have different biases, so it should not come as a surprise that the two primer sets produce different results. But I question the validity to generalize regarding which primer sets that should be considered the 'best' only based on samples from one site.
It is reported that 50-75% of the reads were filtered out due to low quality (if my interpretation of 'net loss' is correct). These are very high numbers (typically these numbers are 10-20%) questioning the quality of the amplicons or sequencing technology (in this particular case). A quality drop around pos 210 is discussed for the V3V4 amplicon (Line 382...). To investigate if this is due to the primer (or specific for the current paper) it would seem natural to include an analyse V3V4 amplicons from other studies.

It is not clear what the comparison between the primers actually shows. E.g. how were normalization done for Fig4? It seems to compare a different number of sequences in each sample. I would say that a more correct way to do this analysis would be to consider the same total number of reads assigned to the listed zOTUs from each primerset.

At best, the data can be used to say that V4V5 is better than V3V4 when analyzing iron oxidizers from the Lo'ihi seamount. But I'm not convinced that the date is even supporting this claim. But somewhat in line with this, the authors conclude that 'it is only through repeated per-environment examinations of primer performance that sturdy foundations for future sudies can be built'.
I guess (with the presented data) a revised paper could be more centered around this claim with the specific aim of evaluating what it the best primer pair to use at Lo'ihi.

Additional comments

The paper lacks clarity and focus.
If the aim of the study is to make a general evaluation of the two primer paris, other datasets should be included in the study.
If the aim is to only make claims about how the two primer pairs 'perform' at Loihi, this should be clearly stated in the abstract, introduction, and discussion.

·

Basic reporting

Smith et al. report a straightforward comparison of environmental microbial diversity surveys generated by two different sets of 16S rRNA primer sets. Both of these primer sets have been commonly used by the community, so the comparison is useful to report. Other studies have performed similar comparisons, but these comparisons have been with simulated communities, and/or with surface seawater or the human microbiome. Smith et al. report the first such comparison with a different kind of habitat - in this case, biofilms from a hydrothermal vent system. The results will be certainly interesting to all hydrothermal vent microbiologists and probably to microbiologists who study other environments as well.

Experimental design

Technical comments:
line 196: "net loss" of reads is confusing without more explanation. Table S1 makes it clear, but a bit more detail would clarify the text. For example: "After quality filtering, denoising, and chimera identification, N reads remained (or were removed). A less important point is that I don't like the word "lost" in this context because it has some negative connotations that I don't think are necessary.

line 201: same comment as above

lines 199 and 204: these percentages apparently refer to the percentages of total original reads? The number of quality-filtered reads not classified at the domain level is much smaller - correct? I think the percentage of quality-filtered reads is the more relevant number because presumably the reads that were already removed prior to taxonomic classification are not "real" reads and not relevant for taxonomic classification anyway.

lines 241-244: I struggled to follow how the different results reported in this one long sentence were connected to each other. I think perhaps each of the (at least three different) results reported here should get its own sentence with a better explanation.

Validity of the findings

The work appears to have been performed well, and it is clearly described, with only a few minor comments below. The conclusions directly follow from the results and are clearly justified.

Raw data has been submitted to NCBI under BioProject PRJNA1080611, but this BioProject does not appear to be publicly available at the time of this review.

---

## Round 0.2 · Minor Revisions

Dear authors,

Thank you for your patience. This is a much improved manuscript, and there are just a few minor revisions that have been recommended by one of the external reviewers. Please address these revisions and we will respond as promptly as practical.

thank you
Pete

Reviewer 1 ·

Basic reporting

No comment

Experimental design

No comment

Validity of the findings

No comment

Additional comments

In the revised paper it is more explicitly stated that the primer comparison is primarily relevant for Kama’ehuakanaloa Seamount (hower future studies may reach similar conclusions for other systems).
I think this improves the paper.

Some comments:

I am still a bit puzzled about the high percentage of reads removed during sequence processing with the DADA2 pipeline. Apparently, this seems to have something to do with a low quality around position 210. I still think this issue deserves some more attention. It would be good to know if this is a general problem with the V3V4 primerset, or something that is specific for this study. In the response to the reviewers, the authors write that: "It would not have been appropriate to include an analysis of other datasets because that would have invited the effect of uncertainty due to differences in methods such as collection & storage, DNA extraction, and sequencing technology."
I don't really understand this argument. It is reasonable to ask if a drop in quality is a general problem with the V3V4 primer set or something that is unique for this study, and the only way to answer it is to compare with other datasets. With the high percentage of reads removed it would also be good to know if the drop in quality represents a bias in some way or the other. Is there a connection beetween sequence and quality (and are some taxa accordingly more likely to be removed than others)? One way to lower the percentage of removed reads may be to crop sequences to 200bp before quality trimming and furhter downstream analyses. It would be interesting to know what impact this has on the results and primer pair evaluation. In any case, it would be good to see a more thoroguh discussion about the possible reasons and impact of the quality drop and the high number of reads removed, and why cropping to 200pb was not done.

I have some problems understanding the reasoning in the disussion about this topic (L388 in revised manuscript):
"The per-base quality of the input sequences varied between primer sets, with the V3V4 reverse sequences showing an abrupt drop in quality around position 210. That drop in quality was not seen in the corresponding V4V5 reverse sequences. The low-quality bases in the V3V4 sequences necessitated much more liberal trimming to maintain acceptable quality scores. This V3V4 trimming may be too exclusive; effectively, because the V4V5 sequences were of higher initial quality, more material remained to be analyzed by the pipeline. This concept is expanded when considering the ratios of completely unclassified taxa between the two primer sets; while the V3V4 primers produced more ASVs, a greater proportion were unidentifiable than those in the V4V5 dataset"

Can you clarify why more 'liberal trimming' is necessary when the quality is low? Why is it a problem that V4V5 were of 'higher initial quality' and what does it mean that "
'more material remained to be analyzed by the pipeline"'? And how is this linked to taxonomic assignments with SILVA?

Please also rephrase "during pipeline" e.g. Line200: "were removed from each single community during the DADA2 pipeline"
Perhaps: "during data processing with the DADA2 pipeline?"

·

Basic reporting

My previous comments have been addressed, and I am satisfied with the revised manuscript.

Experimental design

My previous comments have been addressed, and I am satisfied with the revised manuscript.

Validity of the findings

My previous comments have been addressed, and I am satisfied with the revised manuscript.

---

## Round 0.3 · Minor Revisions

This manuscript is very much improved. The authors have addressed nearly all the matters raised by the two reviewers. In the interest of ensuring that the manuscript is of the highest caliber, I simply ask that the authors speak to this one comment from the reviewer (see below) and resubmit the revision.

I am still a bit puzzled about the high percentage of reads removed during sequence processing with the DADA2 pipeline. Apparently, this seems to have something to do with a low quality around position 210. I still think this issue deserves some more attention. It would be good to know if this is a general problem with the V3V4 primerset, or something that is specific for this study. In the response to the reviewers, the authors write that: "It would not have been appropriate to include an analysis of other datasets because that would have invited the effect of uncertainty due to differences in methods such as collection & storage, DNA extraction, and sequencing technology."

I don't really understand this argument. It is reasonable to ask if a drop in quality is a general problem with the V3V4 primer set or something that is unique for this study, and the only way to answer it is to compare with other datasets. With the high percentage of reads removed it would also be good to know if the drop in quality represents a bias in some way or the other. Is there a connection between sequence and quality (and are some taxa accordingly more likely to be removed than others)? One way to lower the percentage of removed reads may be to crop sequences to 200bp before quality trimming and further downstream analyses. It would be interesting to know what impact this has on the results and primer pair evaluation. In any case, it would be good to see a more thorough discussion about the possible reasons and impact of the quality drop and the high number of reads removed, and why cropping to 200pb was not done.

---

## Round 0.4 · accepted · Accept

Dear authors,

Thank you very much for your patience with this process. This review took longer than I anticipated, and I apologize for the wait. That said I believe your manuscript is much improved and that it will be well received.